# *Bacillus thuringiensis* Cyt Proteins as Enablers of Activity of Cry and Tpp Toxins against *Aedes albopictus*

**DOI:** 10.3390/toxins15030211

**Published:** 2023-03-10

**Authors:** Liliana Lai, Maite Villanueva, Ane Muruzabal-Galarza, Ana Beatriz Fernández, Argine Unzue, Alejandro Toledo-Arana, Primitivo Caballero, Carlos J. Caballero

**Affiliations:** 1Institute of Multidisciplinary Research in Applied Biology-IMAB, Universidad Pública de Navarra, 31192 Mutilva, Spain; 2Departamento de Investigación y Desarrollo, Bioinsectis SL, Plaza Cein 5, Nave A14, 31110 Noáin, Spain; 3Instituto de Agrobiotecnología (IDAB), CSIC-Gobierno de Navarra, 31192 Mutilva, Spain

**Keywords:** *Bacillus thuringiensis*, Cyt toxins, *Aedes albopictus*, synergy, mosquitocidal, Cry toxins, Tpp toxins

## Abstract

*Aedes albopictus* is a species of mosquito, originally from Southeast Asia, that belongs to the Culicidae family and the Dipteran insect order. The distribution of this vector has rapidly changed over the past decade, making most of the temperate territories in the world vulnerable to important human vector-borne diseases such as dengue, yellow fever, zika or chikungunya. *Bacillus thuringiensis* var. *israeliensis* (Bti)-based insecticides represent a realistic alternative to the most common synthetic insecticides for the control of mosquito larvae. However, several studies have revealed emerging resistances to the major Bti Crystal proteins such as Cry4Aa, Cry4Ba and Cry11Aa, making the finding of new toxins necessary to diminish the exposure to the same toxicity factors overtime. Here, we characterized the individual activity of Cyt1Aa, Cry4Aa, Cry4Ba and Cry11Aa against *A. albopictus* and found a new protein, Cyt1A-like, that increases the activity of Cry11Aa more than 20-fold. Additionally, we demonstrated that Cyt1A-like facilitates the activity three new Bti toxins: Cry53-like, Cry56A-like and Tpp36-like. All in all, these results provide alternatives to the currently available Bti products for the control of mosquito populations and position Cyt proteins as enablers of activity for otherwise non-active crystal proteins.

## 1. Introduction

*Aedes (Stegomya) albopictus* (Diptera, Culicidae) (Skuse 1894), commonly known as the Asian tiger mosquito, and originally from Southeast Asia, experimented a quick expansion across the Pacific and Indian Ocean [1], reaching the Americas, Africa and Europe in the past few decades [2]. In Europe, *A*. *albopictus* was first reported in Albania in 1979 and, after its introduction in Italy, it rapidly colonized the rest of the Mediterranean countries [3,4,5]. It is thought that the expansion of this invasive species was mainly favored by the shipment of used tires infested with eggs, which were able to survive until they reached their destination. Additionally, the increase in the temperatures due to climate change may have facilitated its establishment in temperate European countries [6]. Although, *A. albopictus* is generally considered thermophilic; however, because of its ecological plasticity, tolerance to cold temperatures and the ability of its eggs to enter diapause have allowed it to be present in every inhabited continent [7]. For this reason, recent studies predict its establishment in every suitable environment beyond 2050, affecting up to 197 territories of the world [8,9].

Because of its opportunistic blood feeding behavior, *A. albopictus* can be exposed to different pathogens. Although *A. albopictus* can feed on a large number of animals, they prefer humans as a blood source, which results in the infection of more than one billion people with vector-borne diseases every year [10]. This makes it one of the most important vectors of arbovirus diseases, such as chikungunya (CHIKV) [11,12], dengue (DENV) [13] and Zika (ZIKV) [14,15], which are a major threat to public health, as they can cause symptoms like fever, hemorrhages and neurological diseases, among others [16]. In 2007, the first outbreak of chikungunya in Italy causing high fever, joint pain and an itchy skin rash caused public alarm and concern about the reemergence of previously eradicated mosquito-borne diseases in Europe [6].

Today, the control of dipteran vectors of disease is addressed through source reduction, chemical pesticides, biological control, genetic control or combinations of these [17,18]. *A. albopictus* larvae can grow in natural and artificial water containers, either outdoors or in peridomestic environments that can be eliminated when possible or treated with larvicidal insecticides [18]. Larvicide and source reduction are the most efficient methods because they give long-term results. The control of larvae can be achieved through chemical insect growth regulators, organophosphates or *Bacillus thuringiensis* (Bt)*-*based insecticides. Chemicals such as pyrethroids and organophosphates can be highly toxic if used in great amounts, and they are most likely to produce insecticide resistance in mosquitoes [19,20,21]. These have been consistently used as an efficient method to control the spread of mosquito-borne diseases; however, the development of resistances has led to outbreaks and an increased vector competence of mosquitoes [22]. The most common mechanisms of resistance in mosquitoes are genetic mutations on the target sites of the active ingredients or changes in metabolism [21]. 

Bt-based solutions represents a good alternative to chemicals since they have a relatively low environmental impact and a high target specificity that make them eligible for the treatment of drinkable water due to its non-toxicity to humans and animals [23,24]. Bt is a gram-positive and spore-forming bacterium capable of producing a parasporal crystal composed of insecticidal proteins [25]. Strains from the serovar *israeliensis,* Bti, are mainly used for the control of dipteran pests. Their parasporal crystals are mainly composed of four major *δ-*endotoxins (Cry4Aa, Cry4Ba, Cry11Aa and Cyt1Aa), which are highly toxic to insects of the Dipteran order, including mosquitoes. However, recent studies showed that species of the *Aedes* genus had started to develop resistances to the individual Bti proteins Cry4Aa, Cry4Ba and Cry11Aa [26,27,28,29,30]. Among the Bti-based products, Vectobac-*12AS*, a liquid formulation of strain AM 65–52^®^, is probably the most popular [31,32,33]. Although this particular strain had already been evaluated against *A. albopictus* larvae, the effects of each of its individual proteins remained unknown [34]. In this study, we characterized the toxicity of the individual Cry4Aa, Cry4Ba, Cry11Aa and Cyt1Aa proteins against *A. albopictus* larvae. Additionally, we focused on identifying new Cry and Cyt proteins for the control of *A. albopictus* larvae that represent an alternative to the classic Bti toxins and, hence, serve as a preventive measure for upcoming resistances. Finally we found a new Cyt protein, Cyt1A-like, which behaved as a synergistic factor by enhancing the activity of Cry11Aa and as an enabler of activity of three new non-active mosquitocidal proteins.

## 2. Results

### 2.1. Activity of Cry and Cyt Proteins from Strain AM 65-52 against A. albopictus Larvae

In previous studies, the genome sequencing of strain AM 65-52 revealed its pesticidal gene content, which included the *cry4Aa*, *cry4Ba*, *cry10Aa*, *cry11Aa*, *cry60Aa/cry60Ab*, *cyt1Aa*, *cyt2Ba* and *cyt1Ca* genes [35,36,37,38,39]. Since there was no publicly available data on the activity of each of the top four components of its crystals, Cry4Aa, Cry4Ba, Cry11Aa and Cyt1Aa against *A. albopictus* larvae, we decided to evaluate them individually. For this purpose, the genes required for expressing each of the proteins were cloned in vectors optimized for the expression of δ-endotoxins in Bt [40,41], and the resulting plasmids transformed into the acrystalliferous Bt strain BMB171, as previously described [40]. The *cry4Aa* (3543 bp)*, cry4Ba (*3426 bp), *cry11Aa* (1941 bp) and *cyt1Aa* (750 bp) genes were independently cloned, and the resulting recombinant strains expressing *cry4Aa*, *cry4Ba*, c*ry11Aa* and *cyt1Aa* were grown in CCY medium for 48–72 h. The mixtures of spores and crystals were run in an SDS-PAGE. Figure 1 shows that the observed bands are coincidental with the predicted molecular weight of each of the four proteins: 134 kDa for Cry4Aa, 128 kDa for Cry4Ba, 27 kDa for Cyt1Aa and 73 kDa for Cry11Aa.

The mosquitocidal activity of the single *δ-*endotoxins was evaluated on second instar larvae of *A. albopictus* at two different concentrations of spores and crystals (1000 ng/mL and 1.00 ng/mL). Cry4Ba showed the highest activity at 1000 ng/mL, with a mortality of 100%. Conversely, Cry11Aa seemed to be the least active of the tested proteins, with a mortality of 73.33% ± 0.11 at 1000 ng/mL (Appendix A). Once the activity scale for each toxin was defined, the LC_50_ for each of them was calculated, which resulted in 178 ng/mL, 46 ng/mL, 228 ng/mL and 171 ng/mL for Cry4Aa, Cry4Ba, Cry11Aa and Cyt1Aa, respectively (Table 1). Strains AM 65-52 and BMB171 were used as positive and negative controls, respectively. As expected, AM 65-52 showed high mosquitocidal activity, with a LC_50_ of 0.019 ng/mL.

### 2.2. Potential Mosquitocidal Cry, Cyt and Tpp Genes from the BST Collection

To expand the number of potential mosquitocidal proteins with activity against *A. albopictus*, we conducted an in silico screening of 36 Bt wild type strains from our private collection. These were previously isolated from soil samples from several regions and habitats from Spain. Considering that the literature on active Bt proteins against *A. albopictus* larvae is scarce, we used previously described toxins against *Aedes aegypti* (Diptera, Culicidae) and other *Aedes* species as a query to select potential new toxic proteins and synergistic factors [31,33,42,43,44]. Based on our sequencing data, we chose strains BST059.3 and BST-230 as a source of potentiators of toxicity and new mosquitocidal toxins, respectively. Although BST059.3 did not show any activity against *A. albopictus* larvae (LC_50_ > 1 × 10^5^ ng/mL of spore-and-crystal mixture), the strain carried two *cyt1-like* genes: *cyt1A-like* and *cyt1D-like.* Cyt proteins have been extensively described as synergistic factors that are able to potentiate the activity of Bti toxins in different genera of mosquitoes, such as *Aedes*, *Anopheles* and *Culex* [45,46,47]. In particular, Cyt toxins were shown to increase the activity of Cry4Aa, Cry4Ba, Cry10Aa and Cry11Aa when combined together [40,42,45,48]. In agreement with this, we hypothesized that *cyt1A-like* and *cyt1D-like* might be able to produce proteins with a similar function to that of the previously described Cyt toxins. In the case of BST230, several new genes of interest were found, namely, *cry4-like*, *cry53-like*, *cry56-like* and *tpp36-like*. Table 2 contains a list of the new *δ-*endotoxins and their identity percentages with each of their closest matches, including Cyt1Aa5, Cyt1Da1, Cry4Aa4, Cry53Ab1, Cry56Aa2 and Tpp36Aa1.

To study the mosquitocidal properties of the newly found crystal proteins, genes *cry4-like, cry53-like*, *cry56A*-like and *Tpp36*-like were amplified from strain BST-230, cloned into pTBT02, and the resulting vectors electroporated into strain BMB171. In the case of Cry53-like, we were not able to observe any crystals at first (Figure 2A). In order for it to crystallize, we had to include an additional open reading frame (ORF) of 1623 bp downstream of the *cry53-like* coding sequence (CDS) (3004 bp), mimicking its original architecture in the genome. The sequence of this ORF matched the typical Domain V from the crystallization region (C terminal) of Cry1Ac, according to the Conserved Domains Database (CDD) [49]. These C-terminal domain ORFs are often found in mosquitocidal protoxins [50] and, in this case, were required for crystal formation (Figure 2B). Likewise, Cry56A-like also required a fragment of 1692 bp, namely *orf1*, upstream of its CDS (1974 bp) for effective crystal formation (Figure 2B). Analogously, the *cyt1A-*like and *cyt1D-like* genes were amplified from strain BST059.3 and independently cloned into pTBT02. However, neither of them were able to form crystals (Figure 2A). To solve this, both genes were cloned including the *p20* orf, and the resulting plasmids were used to transform strain BMB171. The reason for including *p20* was to promote crystal formation in the Cyt proteins during sporulation (Figure 2B) [51,52].

### 2.3. Characterization of the Bt Recombinant Strains Expressing the New Potential Mosquitocidal Proteins

The BMB171 recombinant strains carrying *cry4-like*, *cry53-like*, *cry56A-like*, c*yt1A-like*, *cyt1D-like* and *tpp36-like* were able to produce spores and crystals when grown in CCY medium for 48–72 h. An SDS-PAGE analysis showed the expected molecular weights for most of the new recombinant proteins. The predicted sizes of Cyt1A-like (~31 kDa), Cyt1D-like (~59kDa) and Cry4-like (~130 kDa) correlated with their observed bands (Figure 3). Cry53-like:orf2 (lane 6) showed two bands: one of ~70 kDa (Cry53-like), which was a little lower than the expected size (76 kDa), and a band of ~62kDa (orf2). For Cry56A-like:orf1 (lane 7), we were able to detect both expected band sizes ~73 kDa (Cry56A-like) and 64 kDa (orf1). Tpp36-like produced a clear band of ~40 kDa (lane 8), although the expected size was about ~55kDa. All of the plasmids were sequenced and did not have any changes and/or mismatches in their sequences.

The mosquitocidal activity of the new *δ-*endotoxins was evaluated on second instar larvae of *A. albopictus*. For these experiments, 1 × 10^5^ ng/mL and 1 × 10^3^ ng/mL of spore-and-crystal mixtures were used as high and low concentrations, respectively. Surprisingly, when analyzing the results, none of the Cry-like and Tpp-like individual proteins showed activity, indicating that they might need to act in partnership with others to produce toxicity.

### 2.4. Synergies between Cyt and AM 65-52 Cry Proteins

To address the potential of Cyt1Aa, Cyt1A-like and Cyt1D-like to produce synergistic interactions with Cry4Aa, Cry4Ba and Cry11Aa against *A. albopictus* larvae, we decided to evaluate the effect of 1:1 mixtures on the aforementioned proteins. The preliminary results of the potential synergistic interactions are shown in Appendix A. A potential synergy was considered when the activity of the 1:1 mix was higher than the sum of the individual activities of the two proteins. Finally, LC_50_ values were calculated. As previously shown in other mosquito species, Cyt1Aa enhanced the activity of Cry4Aa, Cry4Ba and Cry11Aa. However, in the case of the two new Cyt proteins, only Cyt1A-like increased the activity of Cry11Aa, with a synergism factor of 23.14. Cyt1D-like did not potentiate the activity of any of the toxins. Table 3 shows the LC_50_ for each of the binary combinations that produced synergistic interactions.

### 2.5. Cyt Proteins as Enablers of Activity of Cry and Tpp Toxins

Because none of the new potential mosquitocidal proteins found in strain BST-230 showed activity on their own, we wondered if Cyt1Aa, Cyt1A-like and Cyt1D-like might be able to make them active against *A. albopictus* larvae. Considering that Cry56A-like is non-active, we would have expected the LC_50_ of the Cyt1Aa + Cry56A-like mix to equal the LC_50_ of Cyt1Aa (171 ng/mL). However, the activity of the mix was higher by 7.3-fold, with an observed LC_50_ of 23.3 ng/mL. For the Cyt1A-like combinations, the expected LC_50_ was greater than 10^5^ ng/mL since no activity could be observed for them when tested individually. Nevertheless, when mixing Cyt1A-like with Cry53-like, Cry56A-like and Tpp36-like, the LC_50_ values were 1331, 186 and 1053 ng/mL, respectively (Table 4). These results indicated that although Cyt1 proteins are considered synergistic factors of Cry proteins, in some cases, they may behave as enablers of activity of otherwise non-toxic proteins in a specific manner.

## 3. Discussion

Mosquito larvae are highly susceptible to Bti crystals, for which the typical composition is usually a combination of the Cry4Aa, Cry4Ba, Cry11Aa and Cyt1Aa proteins in variable proportions. Interestingly, when analyzing the relative amount of each of the proteins in the parasporal crystal, only a limited number of the anticipated pesticidal proteins are represented and with variable abundance: Cyt1Aa (38–61%), Cry60Ba (5–12%), Cry11Aa (10–27%), Cry4Ba (10–28%), Cry60Aa (2–4%) and Cry4Aa (2–4%) [53]. Although the activity of the major components of the AM 65-52 crystal were previously characterized against *A. aegypti* and species from the genera Culex and Anopheles, little information is available on their effect on *A. albopictus* larvae [54,55,56,57,58]. Additionally, the synergistic interactions between the components of the crystals received attention from different research groups due to the activity of the single Bti proteins being below the toxicity of the complex crystal. Despite this, *A. albopictus* has rarely been used as a model organism for these kind of studies. The reason for this may be the fact that it was traditionally associated with wild animals and territories, and hence, it was a less dangerous species for humans [59]. *A. aegypti*, on the contrary, has often been regarded as the primary vector of arborviruses in human health, becoming the most studied mosquito within its genus. However, in recent times, the *A. albopictus* gained notoriety due to its establishment in Western countries, specifically those in the Mediterranean coast. This led the European authorities to increase the resources allocated to monitoring its spread and to enforce measures to avoid the reemergence of previously eradicated diseases, such as Dengue, especially in countries like Italy and France, where outbreaks of this mosquito have been reported [10,60,61]. For this reason, our research group deemed it appropriate to expand the knowledge on the Bti toxin susceptibility of other Aedes species to *A. albopictus*. We began by characterizing the insecticidal activity of individual Bti toxins on tiger mosquito L2 larvae. The LC_50_ values for Cry4Aa, Cry4Ba, Cry11Aa and Cyt1Aa of 178, 46, 228 and 171 ng/mL, respectively, were similar to the previously described ones for *A. aegypti*. These results suggested that both species may have a similar susceptibility to Bti toxins and crystals [42,44]. When analyzing the synergistic interactions of Cyt1Aa, we found that it was able to potentiate the activity of Cry4Aa, Cry4Ba and Cry11Aa by 28.14-, 32.46- and 11.44-fold, respectively. This synergistic effect on the activity of the individual toxins was considerably higher than previously reported in *A. aegypti*, for which the corresponding synergistic factors were 15.5, 10.91 and 3.15 for the Cyt1A+Cry4A, Cyt1A+Cry4B and Cyt1A+Cry11 combinations, respectively [42,45]. 

Crystal toxins differ in structure and sequence identity, those differences being that Cry proteins are characterized by a three-domain structure [62], Cyt proteins by a single domain constituted by a β-sheet in the middle and surrounded by two α-helical layers [63] and Tpp proteins by a single domain named Toxin_10 (Bin-like) [54]. The mode of action of Cry and Cyt δ-endotoxins is somehow similar at the beginning of the infection, when the crystals are ingested by a susceptible host. Once they reach the midgut, they are solubilized due to the alkaline conditions and processed by proteases into their active form [64]. However, whereas Cyt toxins directly interact with membrane lipids and insert themselves into the membrane of the epithelial host cells, Cry toxins interact with specific receptors of the surface of said cells and oligomerize before the insertion and pore formation occurs [64]. In the case of Tpp proteins, the mode of action is not completely clear, but it was observed that their toxin form binds to the mosquito midgut, specifically the posterior midgut and the gastric caecum [65].

Synergistic interactions between Cry and Cyt toxins have been extensively described in mosquitoes, but the precise mechanism of action remains poorly understood. The most studied combination is the one between Cyt1Aa and Cry11Aa. One of the major assumptions is that Cyt1Aa may function as a receptor for Cry11Aa, facilitating its insertion in the peritrophic membrane of the insect midgut [66]. In agreement with this, Cyt1Aa was shown to delay the development of resistances in mosquitoes by, hypothetically, habilitating new binding sites to other toxins that do not necessarily need to be from the same crystals, such as the *Lysinibacillus sphaericus* Mtx1 and Mtx2 toxins [67]. Currently, there is no evidence of mosquito resistances to Bti crystals as a whole, probably due to the interactions that occur between the proteins within the crystal [68,69].

Although resistance to Bti crystals seems unlikely, results showing the existence of resistant biotypes to single Bti proteins open the possibility of specific populations of mosquitoes becoming resistant to Bti-based solutions over time if used irresponsibly. Therefore, we decided to look for new Bt toxins and synergistic factors that represent an alternative to the most common Bti proteins for the control of *A. albopictus* larvae. Acknowledging that resistances are less likely to develop when synergies take place, one of our focuses was to find new synergistic proteins and characterize them. For this purpose, we selected strain BST059.3, which, despite not showing any activity against *A. albopictus* larvae (LC_50_ > 1 × 10^5^ ng/mL of spore-and-crystal mixture), carried two *cyt1-like* genes: *cyt1A-like* and *cyt1D-like.* In addition, BST230, came across as an interesting strain due to its novel insecticidal content: *cry4-like*, *cry53-like*, *cry56-like* and *tpp36-like*. Cyt1A-like and Cyt1D-like were selected as possible synergistic factors, and the rest of the proteins were selected as hypothetical mosquitocidal proteins. Cry4 toxins have been widely described as one of the major mosquitocidal toxins [54,70,71], whereas, in a previous study, Cry56 showed activity against *A. aegypti* larvae [72]. Despite Cry53 and Tpp36 not been previously reported as potential mosquitocidal toxins, we decided to include them in the study because strain BST-230 showed toxicity against *A. albopictus* larvae (LC_50_ = 39.5 ng/mL) (Appendix A). Interestingly, the Cyt1A-like toxin showed high specificity as a synergistic factor. As opposed to Cyt1Aa, which had activity against *A. albopictus* larvae and interacted synergistically with Cry4Aa, Cry4Ba and Cry11Aa, Cyt1A-like showed no activity (not active at a concentration of 1 × 10^5^ ng/mL) and was only able to potentiate Cry11Aa when tested in combination with the aforementioned proteins. Interestingly, such synergistic effect increased the activity of Cry11Aa 23.14 fold, which was at least two times greater than the one provided by Cyt1Aa (synergistic factor of 11.44).

Possibly the most remarkable feature of Cyt1A-like was the ability to activate the otherwise non-active and newly described Cry53-like, Cry56A-like and Tpp36-like, which showed no activity on their own at a concentration of 1 × 10^5^ ng/mL. This phenomenon was shared with Cyt1Aa for Cry56-like, for which the mixture produced a similar LC_50_. The combination of Cyt1A-like+Cry56A-like was the most effective, with an LC_50_ of 186 ng/mL of spore-and-crystal mixture, a concentration that is close to the ones found in some of the major Bti toxins alone such as Cry4Aa and Cyt1Aa. Cyt1A-like+Cry53-like and Cyt1A-like+Tpp36-like had considerably higher LC_50_ values of 1331 ng/mL and 1053 ng/mL, respectively. Although the combination Cyt1A-like+Tpp36-like was among the least active ones, it confirmed that Cyt1A-like was also able to activate proteins that have a different structure compared to the classic three-domain Cry proteins (Tpp proteins have a typical structure that includes only one domain, named Toxin_10 [54]. In this study, Cyt1A and Cyt1Aa-like were able to activate toxins that were non-active on their own, came from different strains, and had different structures, opening the possibility for them to enable the activity of proteins that have been, until now, considered non-toxic. The biological explanation for this could be that communities of Bt strains in the wild act in a cooperative manner by activating and potentiating the crystal toxins of their fellow Bt neighbors in the pursuit of higher efficacies and more varied mechanisms of actions for when infecting their hosts. Although we were unable to decipher the mechanism of action of said interactions, it may be similar to the one described for Cyt1Aa and Cry11Aa [64]. Here we propose that the characterized Cyt1A and Cyt1A-like toxins could function as enablers of the activity of otherwise non-active Cry and Tpp proteins by habilitating binding sites for them on the lipidic membrane of the insect midgut epithelial cells.

Considering that Cyt1A toxins may delay the appearance of resistance and that the proteins that we described are new, we believe they could represent a great alternative for the control of *A. albopictus* larvae. Additionally, the capability of Cyt proteins to activate non-toxic proteins in a specific manner may be a common characteristic among Cyt1A proteins. To the best of our knowledge, this represents a new function for Cyt proteins, rendering them not only as capable of potentiating the activity of Cry proteins but also as activators of otherwise non-active proteins.

## 4. Materials and Methods

### 4.1. Total DNA Extraction and Genomic Sequencing of the Bacterial Strains

*Bacillus thuringiensis* var. *israeliensis* was isolated from the commercial Bt-based product Vectobac-*12AS^®^*. TF059.3 and BST-230 were obtained from Spanish soils and belong to the BST collection. Total genomic DNA (chromosome and plasmid) was extracted from the strains using the Wizard^®^ Genomic DNA purification kit (Promega, Madison, WI, USA). A sequencing library was prepared for Illumina sequencing by using a NextSeq500 sequencer (Genomics Research Hub Laboratory, School of Biosciences, Cardiff University, Cardiff, UK).

### 4.2. Identification of the Potential Mosquitocidal Genes in the BST Collection

CLC Genomic Workbench 10.1.1 (QIAGEN, Aarhus, Denmark) was used to process and assemble the genomic raw data. Reads were trimmed and filtered to remove those of low quality, and reads shorter than 50 bp were removed. Processed reads were de novo assembled using a stringent criterion of overlap of at least 95 bp of the read and 95% identity, and reads were then mapped back to the contigs for assembly correction. Genes were predicted using GeneMark v2.5 (Georgia Institute of Technology, Atlanta, GA, USA) [73]. To assist the identification process of potential mosquitocidal toxin proteins, the Basic Local Alignment Search Tool [74,75] was deployed against a database built in our laboratory, including the amino acid sequences of known Bt toxins with pesticidal activity from the bacterial pesticidal protein database (https://www.bpprc.org, accessed on 18 July 2022) [76,77]. The pairwise sequence alignment comparison was calculated by using needle v6.6.0 [78]. The prediction of structurally conserved domains was carried out using CD-search [49].

### 4.3. Bacterial Strains and Plasmids Used in the Cloning Process

The recombinant plasmids pHT606:*cry4Aa,* pHT618:*cry4Ba* and pWF45:*cyt1Aa:p20* were provided by Dr. Colin Berry (Cardiff University, Cardiff, UK) and electroporated into the acrystalliferous BMB171 strain. BMB171 was used as a host vector to express all of the proteins used in this study. *Escherichia coli* XL1 blue was used for transformation. *Cry4Aa4-like, cry53Ab1-like, cry56Aa2-like, cyt1Aa5-like, cyt1Da1-like* and *tpp36Aa1-like* were expressed in vector pTBT02. *Cry11Aa* (1941 bp) was cloned alongside *p19* (540 bp) and *p20* (549 bp) [40]. Both P19 and P20 helped crystalize Cry11Aa as well as increase its mosquitocidal activity [79].

### 4.4. Amplification, Cloning and Sequencing of Cyt1a-like, Cyt1d-like, Cry4-like, Cry53-like, Cry56a-like and Tpp36-like

SnapGene^®^ software (GSL Biotech, Chicago, IL, USA) was used to design plasmids and simulate the cloning process.

To clone each of the selected toxin genes, these were first amplified using a specific set of oligonucleotides listed in Table 5.

cyt1-like

Primers harboring the restriction enzymes *PstI and SalI* recognition sites at their extremes were used to amplify the full coding sequence of the *cyt1-like* genes. *SalI* and *SacI* were used for the amplification of *p20.*

cry4-like

The coding sequence of cry4-like was amplified by using primers harboring SalI and SacI recognition sites.

cry53-like

Primers harboring the restriction enzymes SalI and SacI recognition sites at their extremes were used to amplify the full coding sequence of the two contiguous genes cry53-like and orf2.

cry56A-like

Primers harboring the restriction enzymes SalI and SacI recognition sites were used to amplify the gene cry56A-like, whereas PstI and SalI were used for the orf1 gene.

tpp36-like

Primers used for the amplification of tpp36-like harbored SalI and SacI recognition sites.

The PCR reactions were performed using a Q5^®^ High-Fidelity 2X Master Mix (New England Biolabs, Ipswich, SD, USA). PCR products were gel-purified by using NucleoSpin^®^ Gel and a PCR Clean Up kit (Macherey-Nagel Inc., Bethlehem, PA, USA). After the first ligation into pJET-blunt plasmid using a CloneJET PCR Cloning Kit (Thermo Scientific, Waltham, MA, USA), the ligation products were electroporated into *E. coli* XL1 blue cells. Colonies were checked via PCR in order to isolate the ones carrying the plasmid. Plasmids from positive clones were purified using the NucleoSpin^®^ Plasmid Kit (Macherey-Nagel Inc., Bethlehem, PA, USA), following the manufacturer’s instructions. Finally, pJET plasmids sequences were confirmed via sequencing (StabVida, Caparica, Portugal). Once the sequences were verified, the plasmids were digested with the specific set restriction enzymes for each fragment of interest and run in agarose gels, and the corresponding bands were excised and purified. These were then ligated to pre-digested expression vectors using the Rapid DNA ligation kit (ThermoScientific, Vilnius, Lithuania). *Cyt1A-like:p20*, *cyt1D-like:p20*, *cry4-like, cry53-like:orf2*, *orf1:cry56A-like* and *Tpp36-like* were cloned in the pTBT02 vector. The final plasmids were then electroporated into *E. coli* XL1 blue cells. Positive clones were verified via colony-PCR, and plasmids were purified and verified via digestion. *pTBT02-cyt1A-like:p20*, *pTBT02*:*cyt1D-like:p20*, *pTBT02*:*cry4-like*, *pTBT02:cry53-like:orf2*, *pTBT02:orf1*:*cry56A-like* and *pTBT02:Tpp36-like* were finally introduced into the BMB171 Bt strain. The pTBT02 expression vector was created using the pSTAB backbone and by adding a more versatile multicloning site (MCS) as well as a terminator of transcription (TT), which was not present in the previously utilized plasmid. The new MCS and the TT were amplified from the pCN47 plasmid [80] using primers MCS-TT_MfeI and MCS-TT_AatII. The resulting amplicon was cloned in pJET and excised utilizing the MfeI and Aatll enzymes. Next, the digested fragment was inserted in the pSTAB plasmid. Additionally, the *cyt1A* promoter was relocated in pSTAB by amplifying it with new primers carrying the sequence for the Sphl and SalI restriction sites. The resulting fragment was subcloned in pJET, digested using the corresponding restriction enzymes and inserted into the pSTAB.

### 4.5. Nucleotide Sequence Accession Numbers

The nucleotide sequence data reported in this paper were deposited in the GeneBank database under the following accession numbers: OQ397557 for *cyt1A-like*, OQ397558 for *cyt1D-like*, OQ397551 for *cry4-like*, OQ397553 for *Cry53-like*, OQ397554 for *orf2*, OQ397555 for *cry56A-like*, OQ397556 for *orf1* and OQ397552 for *tpp36-like*.

### 4.6. Spore-and-Crystal Mixture Production, Protein Quantification and SDS-PAGE Analysis

BMB171 recombinant strains carrying pHT606:cry4Aa, pHT618:cry4Ba, pWF45:cyt1Aa:p20, pTBT02-cyt1Aa-like:p20, pTBT02:cyt1Da1-like:p20, pTBT02:cry4Aa4-like, pTBT02:cry53Ab1-like:orf2, pTBT02:orf1:cry56Aa-like and pTBT02:Tpp36Aa1-like were grown in 50 mL of CCY medium (supplemented with 20 μg/mL erythromycin) after inoculating single colonies from LB plates [81]. The strains were grown constantly at 28 °C with shaking at 200 rpm. Crystal formation was observed daily at the microscope. Once the cells lysed, after 48–72 h, spore-and-crystal mixtures were washed first with 1M NaCl and 10 mM EDTA, resuspended in 1 mL of dH_2_O water and kept at 4 °C until use. The mixtures were solubilized in carbonate buffer (50 mM Na_2_CO_3_ and 100 mM NaCl, pH 11.3) and quantified for their total amounts of protein by using the Bradford method [82] and by using bovine serum albumin as a standard. For protein profile analysis, the washed spore-and-crystal mixtures were mixed with 2× sample buffer (Bio-Rad, Hercules, CA, USA), boiled at 100 °C for 5 min and then subjected to electrophoresis with a previously described method [83] using Criterion TGX™ 4–20% Precast Gel (Bio-Rad, Laboratories Inc., Hercules, CA, USA). Gels were stained with Coomassie brilliant blue R-250 (Bio-Rad, Laboratories Inc., Hercules, CA, USA) and then destained in a solution of 30% ethanol and 10% acetic acid.

### 4.7. Bioassays on L2 Larvae of A. Albopictus

The toxicities of the single proteins and mixtures were determined on second instar larvae of *A. albopictus.* Eggs of *A. albopictus* were provided by BioGenius GmbH (Friedrich-Ebert-Straße 75, 51429 Bergisch Gladbach, Germany). The bioassays were performed by placing between 10–15 larvae (L2) in each well of a 6-well plate Corning^®^ Costar^®^ (Corning^TM^, Corning, NY, USA). The bioassays were performed following a previously described method [84]. Each well contained a known concentration of spores and crystals in a total volume of 5 mL, with 0.5 mg of brewer’s yeast as food source for the larvae. In order to calculate the median lethal concentration LC_50_, 6 concentrations (from high to low) of Bt suspension were chosen for each recombinant strain: 1000, 500, 250, 125, 62.5 and 31.5 ng/mL for Cry4Aa and Cyt1Aa, 500, 250, 125, 62.5, 31.5 and 15.75 ng/mL for Cry4Ba and 2500, 1250, 625, 312.5, 156.25 and 78.12 ng/mL for Cry11Aa. The highest concentrations (C1) were defined as a dose that produces between 90–100% of mortality, whereas the lower doses were simply 1:2 serial dilutions of the C1 dose.

### 4.8. Statistical Analysis

Concentration–mortality data were subjected to logit regression to estimate the LC_50_ for individual toxins and mixtures of toxins [85].The observed and expected LC_50_ values for the individual toxins and the toxin mixture in *A. albopictus* were used to evaluate the interaction of Cyt1Aa with Cry4Aa, Cry4Ba, Cry11Aa and Cry56Aa2 and the interaction between Cyt1Aa and Cyt1A-like with Cry11Aa, Cry53Ab-like, Cry56Aa-like and Tpp36Ab-like. To calculate the expected LC_50_ values for the toxin mixture under the null hypothesis of no interaction, the “simple similar action” model was used [86]. This model assumes that the concentration–response regression lines for different components of a mixture are parallel and suitable for testing synergism in chemically compounds that are alike, such as Bt toxins. All synergies were evaluated by first calculating the expected LC_50_, as follows, as there were no synergisms between them:$$LC_{50\left(m\right)}\left[\frac{ra}{LC_{50\left(A\right)}}+\frac{rb}{LC_{50\left(B\right)}}\right]\begin{array}{ccc} -1 & & \end{array}$$
where *LC*_50(m)_ is the expected *LC*_50_ of the mixture of toxin A and toxin B, *LC*_50(A)_ is the observed LC_50_ for toxin A alone, LC_50(B) is_ the observed LC_50_ for toxin B alone and rA and rB represent the relative proportions of toxin A and toxin B in the mixture, respectively. All statistical procedures were performed using R software (v.4.1.1) (R Foundation for Statistical Computing, Vienna, Austria).

## Figures and Tables

**Figure 1 toxins-15-00211-f001:**
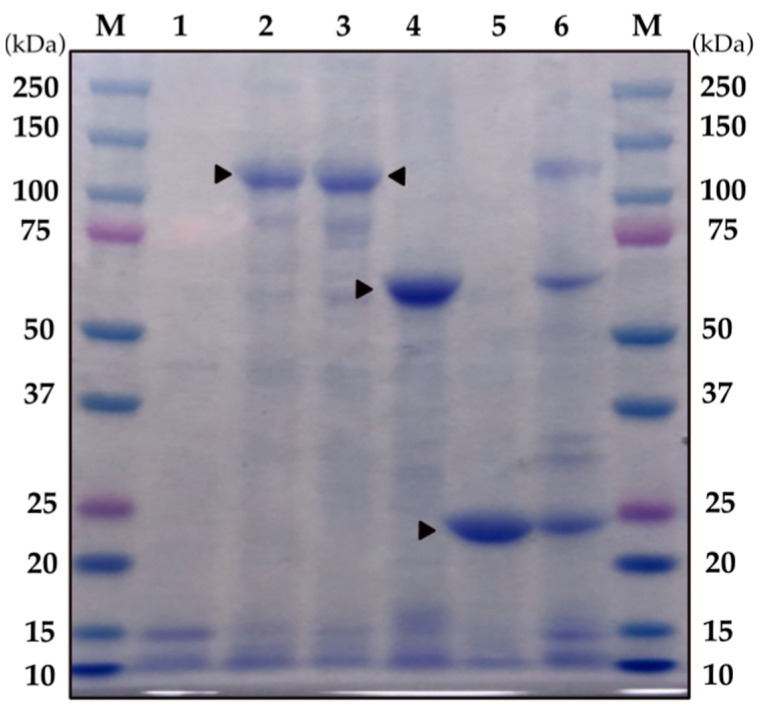
SDS-PAGE showing the protein profile of the BMB171 recombinant strains expressing the AM 65-52 major crystal proteins. Lane M, molecular weight; lane 1, BMB171 carrying an empty plasmid; lane 2, BMB171-Cry4Aa (134 kDa); lane 3, BMB171-Cry4Ba (128 kDa); lane 4, BMB171-Cry11Aa (73 kDa); lane 5, BMB171-Cyt1Aa (27 kDa); lane 6, AM 65-52. Triangles point at major protein bands.

**Figure 2 toxins-15-00211-f002:**
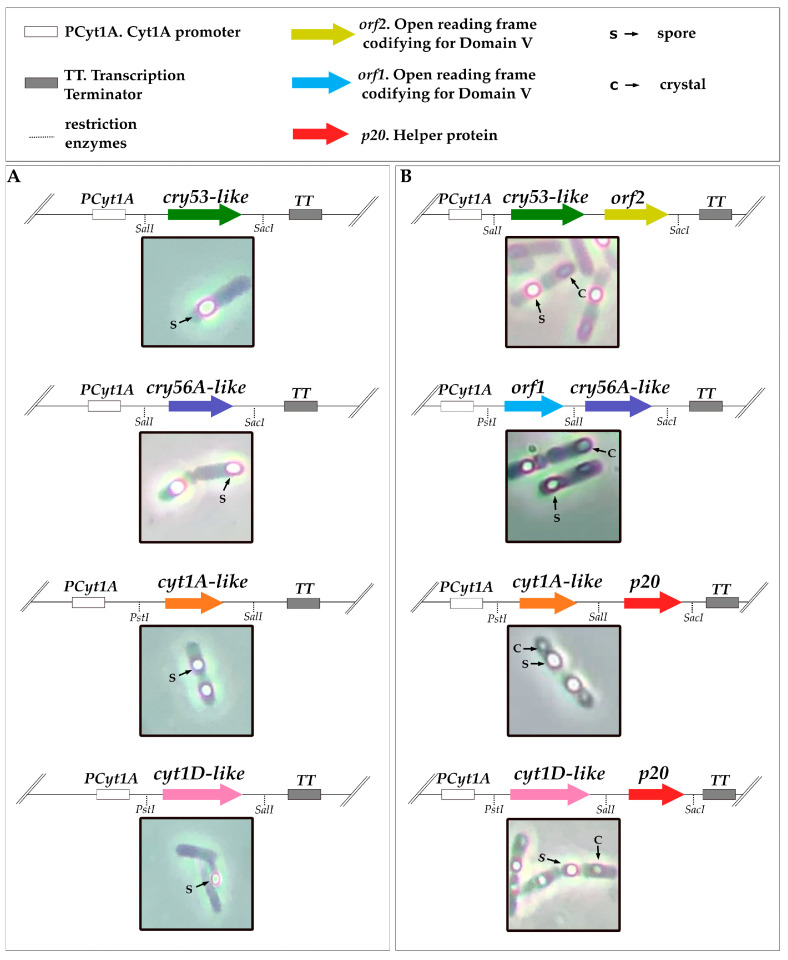
Graphical representation of the impact of helper proteins on the crystallization of Cry53-like, Cry56A-like, Cyt1A-like and Cyt1D-like and of their phenotypes under the microscope after their expression in strain BMB171. (**A**) Images of BMB171 expressing *cry53-like*, *cry56A-like*, *cyt1A-like* and *cyt1D-like* (produce only spores). (**B**) *cry53-like + orf2*, *orf1*:*cry56A-like*, *cyt1A-like:p20* and *cyt1D-like:p20* are able to form both spores and crystals.

**Figure 3 toxins-15-00211-f003:**
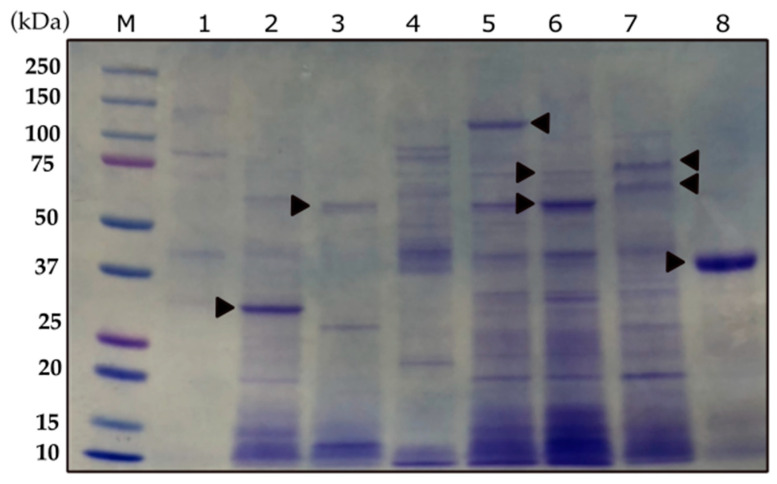
SDS-PAGE showing the protein profile of the BMB171 recombinant strains carrying the new recombinant proteins. Lane M, molecular weight; lane 1, strain TF059.3; lane 2, BMB171-Cyt1A-like (31 kDa); lane 3, BMB171-Cyt1D-like (59 kDa); lane 4, strain BST-230; lane 5, BMB171-Cry4-like (130 kDa); lane 6, BMB171-Cry53-like:orf2 (76 kDa:62 kDa); lane 7, BMB171-orf1:Cry56A-like (64 kDa:73 kDa); lane 8, BMB171-Tpp36-like (55 kDa). Triangles point at major protein bands that correspond with the expressed proteins.

**Table 1 toxins-15-00211-t001:** Mean lethal concentration (LC_50_) value of the AM 65-52 Cry4Aa, Cry4Ba, Cry11Aa and Cyt1Aa proteins for second instar larvae of *A. albopictus*.

Treatment	Observed LC_50_ (ng/mL)	Lower Limits	Upper Limits	χ^2^	df	Slope	SE Slope	Intercept
Cry4Aa	178	142	226	6.66	4	1.80	0.202	−4.04
Cry4Ba	46	30.4	65.7	2.45	4	1.17	0.182	−1.95
Cry11Aa	228	144	324	3.14	4	1.11	0.166	−2.61
Cyt1Aa	171	133	219	4.55	4	1.85	0.223	−4.12
AM 65-52	0.019	0.013	0.024	4.20	4	1.66	0.191	2.90

Treatment: Spore-and-crystal mixtures; LC_50_: median lethal concentration; χ^2^: chi-square; df: degree of freedom; SE: standard error. Control insects experienced no mortality in all cases.

**Table 2 toxins-15-00211-t002:** List of new *cry* and *cyt* genes selected for this study.

Target Database	Pairwise Identity %	MW (kDa)	Accession Number of Reference	Accession Number	Strain
*cyt1Aa5*	65	31	CAD30079	OQ397557	BST059.3
*cyt1Da1*	48	59	ADV33305	OQ397558	BST059.3
*cry4Aa4*	38	132	AFB18317	OQ397551	BST230
*cry53Ab1*	40	76	ACP43734	OQ397553	BST230
*cry56Aa2*	54	73	ADK38584	OQ397555	BST230
*tpp36Aa1*	30	56	AAK64558	OQ397552	BST230

MW: Molecular weight.

**Table 3 toxins-15-00211-t003:** Mean lethal concentration (LC_50_) value of the 1:1 mixture calculated for L2 larvae of *A. albopictus*.

Treatment	Observed LC_50_ (ng/mL)	Expected LC_50_ (ng/mL)	Lower Limits	Upper Limits	χ^2^	df	Slope	SE Slope	Intercept	Synergism Factor
Cyt1Aa + Cry4Aa	6.04	170	4.68	8.13	1.91	4	1.66	0.212	−1.30	28.14
Cyt1Aa + Cry4Ba	3.08	100	2.05	5.29	7.42	4	2.14	0.243	−1.05	32.46
Cyt1Aa + Cry11Aa	17.1	195.6	13.8	21.0	1.92	4	2.37	0.257	−2.93	11.44
Cyt1A-like + Cry11Aa	19.7	456	16.1	24	0.704	4	2.04	0.215	−2.69	23.14

Treatment: Spore-and-crystal mixtures; Expected LC_50_: Expected median lethal concentration calculated with the method Tabashnik (1992); χ^2^: chi-square; df: degree of freedom; SE: standard error; Synergism Factor: the ratio of the expected LC_50_ and the observed LC_50_. Control insects experienced no mortality in all cases.

**Table 4 toxins-15-00211-t004:** Mean lethal concentration (LC_50_) value of the 1:1 mixture (enabler+Cry/Tpp) and the toxins inoculated individually, calculated for L2 larvae of *A. albopictus*.

Treatment	Observed LC50 (ng/mL)	Lower Limits	Upper Limits	χ^2^	df	Slope	SE Slsope	Intercept
Cyt1Aa	171	133	219	4.55	4	1.85	0.223	−4.12
Cry56A-like	>10^5^							
Cyt1Aa+Cry56A-like	23.3	19.2	27.9	3.71	4	2.20	0.220	−3.01
Cyt1A-like	>10^5^							
Cry53-like	>10^5^							
Cyt1A-like+Cry53-like	1331	1091	1649	2.87	4	2.03	0.202	−6.35
Cyt1A-like	>10^5^							
Cry56A-like	>10^5^							
Cyt1A-like+Cry56A-like	186	138	239	6.26	4	1.59	0.192	−3.60
Cyt1A-like	>10^5^							
Tpp36-like	>10^5^							
Cyt1A-like+Tpp36-like	1053	860	1298	5.85	4	2.04	0.208	−6.18

Treatment: Spore-and-crystal mixtures; χ^2^: chi-square; df: degree of freedom; SE: standard error. Control insects experienced no mortality in all cases.

**Table 5 toxins-15-00211-t005:** Primers used for PCR and sequencing.

Primer Name	Sequence (5′-3′)
Cyt1Aa_like_FW_PstI	GTGTCGACCAAAGGCAGTGGTGTTTTAAG
Cyt1Aa_like_RV_SalI	CTCTGCAGGGGCTACCCAATTATAATCG
p20-Fw-SalI	CCTGCAGGGATAAAATTGGAGGATAATTGATG
p20-Rv-SacI	GGCATGCGTTTCCAGTGCATTCAATTTAC
Cry4Aa4_FW_SalI_BST230	GTCGACGAAATTCAATTGGAAATGGAGGAAC
Cry4Aa_RV_SacI_BST230	GAGCTCCTTTTTTCCAAATTTGTAATAGAAT
Orf_Cry56_FW_PstI	CTGCAGCAGCAAAAAATACGCAGAAAAGGTA
Orf_Cry56_RV_SalI	GTCGACGAATCGTTAACGGTTATATCTTTG
Cry56Aa2_FW_SalI_BST230	GTCGACGGACTACATAAGGAGTGAAA
Cry56Aa2_RV_SacI_BST230	GAGCTCCTATAGAACTGGCCGCTTGA
Cry53+Orf2_FW_SalI	GTCGACGGACTACATAAGGAGTGAAAAAT
Cry53+Orf2_RV_SacI	GAGCTCCTAATTCTCATTTGGAATCGT
Tpp36Aa1_FW_SalI_BST230	GTCGACGAAAAAAATCACATAAGGAGTG
Tpp36Aa1_RV_SacI_BST230	GAGCTCCCCTTACTTCGTTCTACTTAC
Cyt1Aa4like_FW_PstI	CTGCAGCAAAGGCAGTGGTGTTTTAAG
Cyt1Aa4Like_RV_SalI	GTCGACGGGCTACCCAATTATAATCG
Cyt1Da1_like_FW_PstI	CTGCAGCGAGAGAGGTATAAATATGAACC
Cyt1Da1_like_RV_SaII	GTCGACGTAAGAACCCTACGACTAGG
MCS-TT_MfeI	CAATTGGCATGCCTGCAGGTCGACTCTAGAAGATCTCCCGGGTACCGAGC
MCS-TT_AatII	GACGTCAAAGGCGCCTGTCACTTTGCTTG
Sequencing	
Cry4Aa_seq_BST230	CTAGTGAATAATGTAGGTTCTTTA
Cry4Aa_seq1_BST230	CAAGTATGCAATACTGCTTAC
Cry4Aa_seq2_BST230	GATATGGTTTCTATTTCACTTG
Cry4Aa_seq3_BST230	GTCAATCAAGAAATTTACTTCAAA
Cry56orf_seq_FW	GAAGTGTCACGATCGCCAT
Cry56orf_seq_RV	TTCACATGTTCCAATGCTTCA
Cry56orf_seq_FW_1	ATTCCGGCTGCACATGTAAC
Cry56orf_seq_RV_1	GAGCTGTTTGGTGAAGTATCCA
Cry56orf_seq_FW_2	CCATAACATTATATACTAACGTGG
Cry56orf_seq_RV_2	TACTGCTCAGATGCCACGTT
Cry53like_FW_seq1	GTAGAGAAATGACCATAACAG
Cry53like_RV_seq2	GCAGGAAATAGAGCAACTATATCT
Cry53like_FW_seq3	GCTTTGTCACTAAATAATTTGCG
Cry53like_RV_seq4	GTAAGCAAAATTCTCATTTCGCAA
Cry53like_RV_seq5	CATACCTAAGTTTGTATTTGTATCT
Cry53like_FW_seq6	GATTTTCATATTGACACAGGAGA
Tpp36like_FW_seq1	CATTAATTCCGTGTATACTTGTAAA
Tpp36like_RV_seq2	CTGCTAATGAATATTGATAATCA
Seq pCyt1A F (59)	CATATATTTGCACCGTCTAATGG
MCS-TT_AatII	GACGTCAAAGGCGCCTGTCACTTTGCTTG

Underlined nucleotides represent restriction enzyme sites.

## Data Availability

Not applicable.

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
