# Peer review of "Bacillus thuringiensis Cyt Proteins as Enablers of Activity of Cry and Tpp Toxins against Aedes albopictus"

_toxins, 2023, doi:10.3390/toxins15030211_

Round 1

Reviewer 1 Report

Introduction

The introduction is well structured and provides sufficient information for readers to understand the aim of the study.

Results

p 2, line 87ff: Do you mean “anticipated pesticidal protein”?

Does the B. thuringiensis strain AM 65-52 express all the analysed proteins? If yes, what is your explanation for the missing protein band of Cry43A in Figure 1.

I personally think Tables 1 and 2 could be combined by including the observed LC50 in table 1, the other data might be more suited for the supplementary. In general, the tables contain a lot of information that might not be directly necessary to understand the results.

The legend of Figure 3 states, “triangles point at major hypothetical protein bands” but the protein bands are not hypothetical themselves.

The subsection 2.4 concerning the synergistic effect is quite hard to follow.

Discussion

The discussion is rather short and is somewhat a repetition of the results. I also do not see the added value of Figure 4. The authors could omit this figure and explain the proposed mechanisms in more detail in the discussion.

Material & Methods

The M&M section is described well, however the section regarding the cloning of the different toxin variants is a bit hard to read/understand. I miss some more information about the different endotoxins or the Tpp proteins. Throughout the whole manuscript, I was looking for the difference between Tpp proteins and Cry/Cyt proteins (or if there is any).

Minor Points

Inconsistent spelling of restriction enzymes, protein/gene and strain names (cursive or not cursive, uppercase vs. lowercase); references are partially not converted to numbers. For example, you jump between Bti and Bt as an abbreviation for B. thuringiensis, and it is unclear why the variant israeliensis was used or if it was used in all the experiments.

LC50 is often written as “CL50” and once as “LC10” (I did not understand if the latter was intentional).

p 2, line 86: “and”       
p 12 l 349 seems to miss a word
p12 l 387: bracket missing       
Table 6 legend: “restriction enzyme sites” or something similar 
p14 l 428: “using serum albumin” as a standard?          
p14 l 437f: This sentence might be missing a word or should be checked for grammatical errors       

Reviewer 2 Report

The research work is good, but the manuscript needs substantial improvement in the Results and Material and method section. The population collection and rearing are completely missing in the manuscript. Results need concise and only present the current tables and figures. Discussion should be improved. My comments and suggestions are given in the PDF file attached. The manuscript can be considered with major revision.

Reviewer 3 Report

The MS presented shows interesting data on Bacillus thuringiensis toxins with activity against Aedes albopictus. Synergistic effect between Cry and Cyt toxins from B. thuringiensis subsp. israelesis have been studied extensively in mosquitoes A. aegypti, and here the authors showed that these toxins have also similar synergistic effect on A. albopictus. However, new Cyt-like and Cry-like toxins were studied and interestingly presented larvicidal activity against A. albopictus solely when Cyt-like and Cry-like were mixed. Tpp-like and Cyt-like toxins showed the same behavior. In my opinion the MS should be published after authors address minor points that I list below:

1.     Lines 234-235: “To test this each of the Cyt1Aa, Cyt1A-like and Cyt1D-like were mixed with the Cry4-like, Cry53-like, Cry56A-like and Tpp36-like proteins”.

Data of larvicidal activity are missing in table 5: between Cyt1A+Cry53-like, Cyt1A+Tpp36-like; and data of Cyt1D-like with Cry4-like, Cry53-like, Cry56A-like.

2.     Lines 274-278: “Surprisingly, this synergistic effect on the activity of the individual toxins was considerably higher than previously reported in Aedes aegypti, for which the susceptibility of the binary mixtures between Cyt1Aa and the three major Bti Cry toxins was not as effective, with synergistic factors of 15.5, 10.91 and 3.15 for the combinations Cyt1A+Cry4A, Cyt1A+Cry4B and Cyt1A+Cry11, respectively”.  

As far I know, activity assay in A. aegypti are performed in larvae L4, while in this work were used L2. Insect sensitivity to B. thuringiensis toxins is higher in early stage, so this phenomenon should be discussed.

3.     Finally, I am not completely agreeing with the concept of “activator” for Cyt toxins. The proposed model in figure 4 agrees with ref 71, where Cyt toxin could be a kind of “receptor”. However, synergistic effect also is unsuitable due none effect was observed when toxicity was tested with individual toxins. I recommend using an alternative concept more appropriated.  

Round 2

Reviewer 2 Report

The manuscript is much improved and can be accepted for publication.